# Differences in ligand-induced protein dynamics extracted from an unsupervised deep learning approach correlate with protein–ligand binding affinities

Ikki Yasuda[1], Katsuhiro Endo[1], Eiji Yamamoto [2], Yoshinori Hirano[1,3] & Kenji Yasuoka [1✉]

Prediction of protein–ligand binding affinity is a major goal in drug discovery. Generally, free energy gap is calculated between two states (e.g., ligand binding and unbinding). The energy gap implicitly includes the effects of changes in protein dynamics induced by ligand binding. However, the relationship between protein dynamics and binding affinity remains unclear. Here, we propose a method that represents ligand-binding-induced protein behavioral change with a simple feature that can be used to predict protein–ligand affinity. From unbiased molecular simulation data, an unsupervised deep learning method measures the differences in protein dynamics at a ligand-binding site depending on the bound ligands. A dimension reduction method extracts a dynamic feature that strongly correlates to the binding affinities. Moreover, the residues that play important roles in protein–ligand interactions are specified based on their contribution to the differences. These results indicate the potential for binding dynamics-based drug discovery.

[1] Department of Mechanical Engineering, Keio University, Yokohama, Kanagawa, Japan. [2] Department of System Design Engineering, Keio Univsity, Yokohama, Kanagawa, Japan. [3] Laboratory for Computational Molecular Design, RIKEN Center for Biosystems Dynamics Research (BDR), Suita, Osaka, Japan. ✉email: yasuoka@mech.keio.ac.jp

In computational drug discovery, the estimation of binding affinities between the target proteins and ligands is one of the main goals. Various approaches have been proposed and performed for both physics-based and data-driven methods. In physics-based approaches, protein–ligand free energy calculations have been widely conducted using free energy perturbation and thermal integration methods, and the results agree well with experimental data[1–5]. However, despite the high accuracy, the high-calculation cost has prevented its practical use[6]. Data-driven approaches, such as scoring functions for docking, quantitative structure-activity relationship method with machine learning, and deep learning methods have been studied over the past few decades[7,8]. Deep learning approaches can grasp important characteristics automatically from the high-dimensional data of proteins and ligands. The approaches for protein–ligand affinity prediction have succeeded in finding relevant patterns in 3D structures[9–12] and protein and ligand sequences[13,14] using supervised learning with a sufficient amount of dataset. Although there are widely used databases such as PDB-bind[15] and DUD-E[16] for protein–ligand-binding data, an efficient approach cannot be determined if the available dataset is limited[17].

Protein dynamics play an important role in biological phenomena. All-atom molecular dynamics (MD) simulations are powerful tools that can generate a large amount of dynamic data and analyze protein dynamics at the atomic level, along with experimentation[18,19] and coarse-grained simulations[20]. MD data analysis for protein dynamics has focused on protein fluctuations, relaxation time, stability, and state transitions. The commonly used methods are root mean square deviation, principal component analysis, relaxation time analysis, decomposition cross-correlation maps, and root mean square fluctuation (RMSF)[21–24]. For protein and ligand systems, these methods have revealed that protein dynamics changes before and after ligand binding[25–28]. Recently, machine learning has been combined with MD to utilize the vast amount of MD data[29]. In particular, several machine-learning-assisted methods succeeded in extracting important molecular dynamics[30–33], and were applied to complex biomolecules[34–36]. For instance, VAMPnets[34] demonstrated the kinetics of metastable states of protein folding and unfolding by Markov states models using deep neural networks (DNNs) instead of traditional handcrafted procedures. Tsuchiya et al. compared the time-series trajectories of a protein with or without a bound ligand using an autoencoder to automatically detect the allosteric dynamics[35].

Although the various methods for the MD data analysis could identify changes in ligand-induced protein dynamics, the link between dynamics and ligand affinity is not fully understood. It has been experimentally investigated that the large conformational change at the binding pocket such as transition between open and closed states of the binding pocket[37,38], and ligand-induced local secondary structure change[28], is related to the ligand binding affinity. However, it is challenging to estimate the ligand binding affinity from just the subtle change included by the short-term MD trajectories[39].

In this study, we propose a method to predict binding energies from subtle change in proteins dynamics upon ligand binding, using a deep learning approach for MD data analysis[31]. In contrast to general approaches using descriptors and supervised machine learning[39,40], our method uses raw MD trajectories of a ligand binding site with different kinds of ligands, and quantitatively measures the differences in the dynamics using unsupervised learning. The method performs (1) dimension reduction for the dynamics feature and, (2) detection of the residues whose dynamics significantly changed due to interaction with the ligands. We verified the method in two systems, bromodomain 4 (BRD4)[41,42] and protein tyrosine phosphatase 1B (PTP1B)[43,44]

systems, which have been used for benchmark of free energy calculation in previous studies[1–5]. We indicate a strong correlation between the extracted feature and binding energies and suggest that the feature relevant to dynamics can work as a predictor of binding energy. In addition, the significant dynamics change in the detected residues dictate potential binding between the residues and the ligand.

## Results

**Unsupervised learning for ligand-induced dynamics.** Here, we present unsupervised learning procedures to extract features of protein dynamics and detect residues whose dynamics is highly influenced by ligand binding (see details in the Method section). Firstly, to represent protein dynamics, we use local dynamics ensemble (LDE) that is obtained from MD simulations (Fig. 1a, b)[31]. The LDE is defined as an ensemble of short-term trajectories $x$ of particles of interest, i.e., the binding site residues. We assume that the local dynamics is affected by ligand interactions, therefore it is related to the ligand affinities. Then, the differences in the LDE distributions between ligand-binding or ligand-free systems (Fig. 1c) are measured based on Wasserstein distance[45,46],

$$W_{ij} = \mathbb{E}_{x \sim y_i}\left[f_{ij}^*(x)\right] - \mathbb{E}_{x \sim y_j}\left[f_{ij}^*(x)\right], \qquad (1)$$

where $\mathbb{E}$ is expectation over the probability distribution, the lower indexes $i, j$ are the systems and $y_i$ is the probability distribution of the LDE of system $i$. We approximate the optimal function $f_{ij}^*$ by DNNs.

The Wasserstein distance is calculated for all pairs of $N$ ligand systems, resulting in a distance matrix of $(N, N)$ (Fig. 1c). The high dimension of the distance matrix makes it difficult to extract simple features based on understanding the global differences in systems. Therefore, the distance matrix is embedded into low-dimensional $N$ vectors that represent the systems using a non-linear dimension reduction. Then, the first and second principal components are extracted using principal component analysis. We evaluate the extracted variables by referring to their correlation to ligand-binding affinities (Fig. 1d and Supplementary Fig. 1 for the workflow).

In the other branch, we interpret the differences of protein dynamics using a function $g_{ij}(x_i)$,

$$g_{ij}(x_i) = \mathbb{E}_{x \sim y_j}\left[f_{ij}^*(x_i) - f_{ij}^*(x)\right], \qquad (2)$$

that shows how each short-term trajectory in system $i$ differs from the average dynamics of system $j$ (Fig. 1c and Supplementary Fig. 2). Since $g_{ij}(x_i)$ is obtained for short-term trajectory that includes multiple residues, we could further specify the residues which are highly related to the Wasserstein distance (Fig. 1e). According to $g_{ij}(x_i)$, we classify the short-term trajectories of system $i$ into system-$i$-characteristic and system-$j$-similar groups,

$$x_i \in \begin{cases} X_{ij}^C, & \text{if } g_{ij}^C \leq g_{ij}(x_i) \\ X_{ij}^S, & \text{if } g_{ij}(x_i) \leq g_{ij}^S \end{cases} \qquad (3)$$

where $X_{ij}^C, X_{ij}^S$ are the system-$i$-characteristic and system-$j$-similar groups, and $g_{ij}^C, g_{ij}^S$ are the higher and lower thresholds. Here, we set the $g_{ij}^C$ and $g_{ij}^S$ to the boundaries of the highest and lowest 10% of all the sampled $g_{ij}(x_i)$. To clarify the specific dynamics contributing to the $W_{ij}$, we examine the residues included in the LDE by comparing the $X_{ij}^C, X_{ij}^S$ groups, i.e., characteristic or non-characteristic dynamics of system $i$. We introduce a physical property to represent short-term trajectories and verify the property's correspondence to the distinction of $X_{ij}^C, X_{ij}^S$ groups.

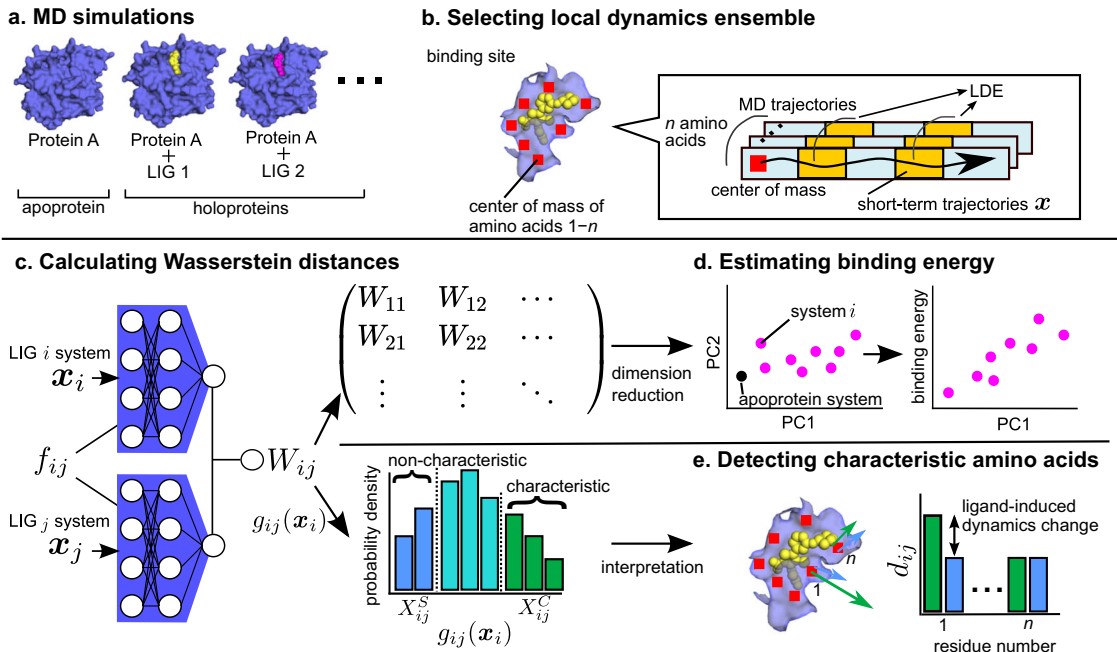

**Fig. 1 Workflow to detect differences in ligand-induced protein dynamics. a** Molecular dynamics (MD) simulations for ligand-free (apoprotein) and ligand-bound (holoprotein) systems. **b** Ligand-induced protein dynamics is represented by local dynamics ensemble (LDE), which is an ensemble of short-term trajectories **x** of the center of mass of n binding-site residues. **c** The difference of the LDEs between system i and j is calculated based on Wasserstein distance $W_{ij}$ using deep neural networks (DNNs) $f_{ij}$. The Wasserstein distances are calculated for all pairs of systems and the distance matrix is obtained. In addition, each short-term trajectory in system i is represented by the output of the DNNs which is denoted as a function $g_{ij}(\boldsymbol{x}_i)$. In histogram bottom, high $g_{ij}(\boldsymbol{x}_i)$ (green) and low $g_{ij}(\boldsymbol{x}_i)$ (blue) indicate that the short-term trajectories are characteristic to system i and similar to system j, e.g., characteristic to apoprotein and similar to holoprotein, respectively. **d** The matrix of Wasserstein distances is embedded into points in a lower-dimensional space, and principle component analysis is performed to the embedded points. The first principal component (PC1) is compared to ligand-binding energies. **e** The difference detected by $g_{ij}(\boldsymbol{x}_i)$ is interpreted, and the residues whose dynamics are changed by ligand interactions are examined. For the characteristic and non-characteristic trajectories, short-term mean square displacement (RMSD) $d_{ij}$ is calculated per residue. If the large gaps of $d_{ij}$ are observed between the characteristic and non-characteristic trajectories, the residues are highly influenced by the ligand.

Here, short-term root-mean-square displacement (RMSD) was calculated for each residue included in the LDE,

$$d_{ij}(n) = \frac{1}{N_{ij}} \sum_{\boldsymbol{x}_i \in X_{ij}} \left[ \frac{1}{T - T_0} \sum_{\Delta = T_0}^{T} \| \boldsymbol{r}_n(t + \Delta) - \boldsymbol{r}_n(t) \| \right], \quad (4)$$

where n is the index for the resides in LDE, $N_{ij}$ is the number of short-term trajectories in $X_{ij}$, T is the time of LDE, $T_0$ is time when the short-term RMSD converged to plateau (Supplementary Fig. 3), $\boldsymbol{r}(t)$ is the first frame of $\boldsymbol{x}_i$ on the time t in MD simulations. If the short-term RMSDs between $X_{ij}^C$ and $X_{ij}^S$ are distinct, i.e., $d_{ij}^C(n) \ll d_{ij}^S(n)$ or $d_{ij}^C(n) \gg d_{ij}^S(n)$, the dynamics of the residues contribute to $W_{ij}$.

**LDE of protein–ligand systems**. We performed a total of 13.2-μs all-atom MD simulations [10 ligand-bound (holo) protein and one ligand-unbound (apo) protein system] to obtain trajectories of the BRD4 systems (Fig. 2). Three 400 ns independent production runs were executed with different initial velocities for each system. The initial structures of the complexes and the stability of simulations are shown in the Supplementary material (Supplementary Figs. 4–6).

The features of protein dynamics from the MD simulation data were represented using the LDE. The LDE should have an appropriate selection of particles and time to contain important dynamics of interest. For the particle selection, we assume that the behavior of amino acids is sensitive to the presence of a ligand, i.e., the binding site shows representative dynamics that are induced by ligand interactions. We note that this selection

includes no information on the bound ligand, making it possible to directly compare the behavior of the ligand-binding sites between systems. As for the time, we selected a very short time frame (128 ps). Interestingly, the protein dynamics in this short scale varied depending on the species of the binding ligand (Supplementary Fig. 7). This time scale typically corresponds to local dynamics of side-chains.

The properties of the LDE with the selection were analyzed using the function $g(\boldsymbol{x})$. The $g(\boldsymbol{x})$ were distributed similarly regardless of the initial conditions (Supplementary Fig. 8). Moreover, the local dynamics were distributed evenly throughout the MD simulations (Supplementary Fig. 9), showing that they were not influenced by the slow fluctuations. These results indicated that the local dynamics were robust with respect to the differences in the initial conditions and long-term dynamics.

**Feature for protein dynamics correlates with binding affinity**. The differences of ligand-induced dynamics in the systems were calculated based on the Wasserstein distances of the LDEs (see Eq. (1)). The distance matrix indicates that apoprotein system $S_0$ is separated by a relatively larger distance from the holoprotein system than that between the holoprotein and another holoprotein system (Fig. 3a). The distance embedding demonstrates clear differences in protein's short-term dynamics in the systems (Fig. 3b). As shown in the distance matrix, the apoprotein system $S_0$ is separated from the holoproteins. Moreover, systems with lower-affinity ligands tend to position near apoprotein compared to systems with higher affinity ligands. The link between the first

principal component (PC1) and binding affinities was quantitatively evaluated by comparing it to the binding energies calculated in a previous study[2]. Pearson's product moment correlation coefficient between PC1 and the binding energies was 0.88 (Fig. 3c).

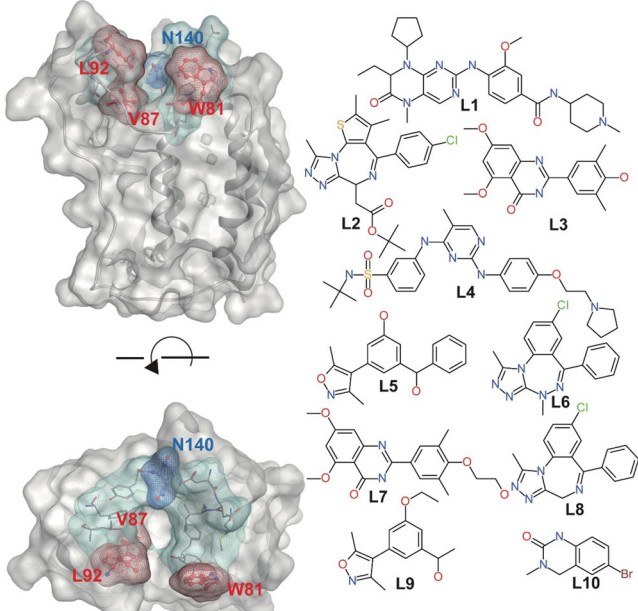

**Fig. 2 Bromodomain 4 (BRD4) system.** Molecular surface of BRD4 (Protein Data Bank ID: 2OSS) superimposed on ribbon diagram (left column). The colored (green, red, and blue) meshed molecular surfaces indicate the ligand binding site. The key residues (Pro81, Val87, and Leu92) and not detected residue (Asn140) are shown in red and blue meshed molecular surface on ball and stick models with labels, respectively. Chemical structures of ligands L1–L10 (right column). The L1 to L10 labels correspond to the ligands of the 1 to 10 holoprotein systems.

**Residue-level interpretation on ligand-induced dynamics.** Since the feature of dynamic differences, i.e., PC1, indicates ligand affinity, the interpretation of "difference of dynamics", i.e., Wasserstein distance, provides insights into the mechanisms of ligand binding, ligand interactions, and protein stabilization. We examined the difference in dynamics in holo- and apoprotein systems to find which amino acids were most influenced by the ligand.

In particular, we compared the apoprotein system to the ligand 3 (RVX-OH) system. Since the ligand 3 system was most distant on PC1 in Fig. 3b, the dynamics difference is most clear in the ligand 3 and apoprotein systems. Here, characteristic dynamics to the apoprotein, i.e., apoprotein-like, were detected using $g(\boldsymbol{x})$ (see Eq. (2)) and characteristic groups of the trajectories, and the characteristic behavior was clarified using short-term RMSD for the residues (see Eq. (4)). Figure 3d compares flexibility for residues between the three $g(\boldsymbol{x})$ groups, $X_{ij}^C$, $X_{ij}^M$ and $X_{ij}^S$. Here, $X_{ij}^M$ is the middle of $X_{ij}^C$ and $X_{ij}^S$, i.e., the trajectories in $X_{ij}^M$ meet $g_{ij}^S < g_{ij}(\boldsymbol{x}_i) < g_{ij}^C$. The characteristic behavior in the apoprotein system $X_{ij}^C$ demonstrated large movements in all amino acids, indicating that the apoprotein was more flexible than the holoprotein. The most distinct differences between the groups are found in Trp81, Val87, and Leu92. Therefore, we concluded that these residues are most influenced by the ligand, i.e., the key residues in ligand binding. We estimate that the detected residues whose dynamics vastly changed by ligand have important roles in the ligand binding. To verify this, we refer to experimental and other computational literature on BRD4. Ligand-induced dynamics changes on Trp81 were observed by nuclear magnetic resonance[47]. This study showed the change in dynamics of Trp81 correlated to the ligand-binding affinity, in agreement with our result. In addition to Trp81, our result suggests that the other two resides, which have not been experimentally identified yet, have important roles in the ligand-binding. Previous simulation studies suggested that the residues at the binding site can make hydrophobic interactions[48,49] and are expected to contribute to ligand binding. Although these two residues are not

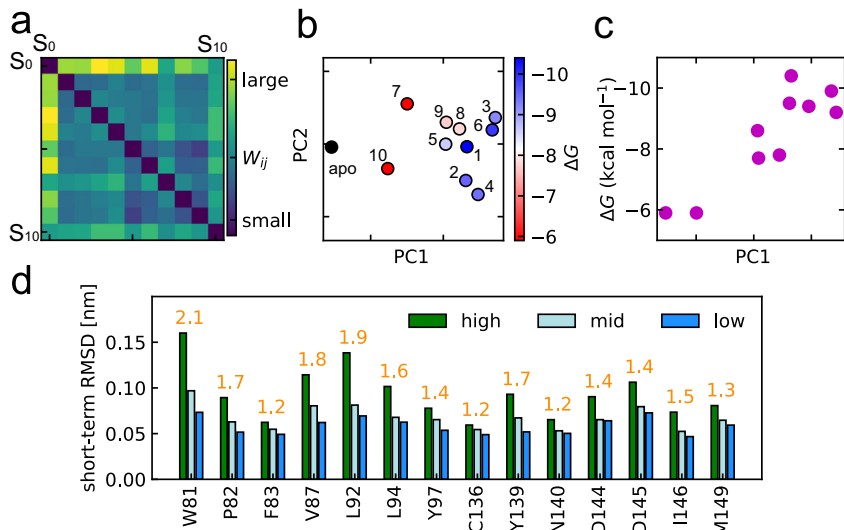

**Fig. 3 Differences in the ligand-induced protein dynamics of the BRD4 systems. a** Distance matrix of Wasserstein distances between probability distributions of the LDEs from system 0–10. Sytem 0 is the apoprotein system and others are labeled according to the ligand. The large Wasserstein distance (yellow) corresponds to a large difference in the protein dynamics. **b** Embedded points of the distance matrix. The points corresponds to the systems which are colored according to the binding energies and the apoprotein system is in black. The binding energies were obtained from a previous computational research[2]. **c** Correlation between PC1 and the binding energies. **d** Characteristic dynamics to the apoprotein system was interpreted for the binding-site residues, in comparison to ligand system 3. The short-term trajectories of the apoprotein system were classified into apo-characteristic (high), holo-like (low), others (mid) groups, and short-term RMSD $d_{ij}$ was calculated.

 COMMUNICATIONS BIOLOGY | (2022)5:481 | https://doi.org/10.1038/s42003-022-03416-7 | www.nature.com/commsbio

experimentally validated, we hope further experimental research will demonstrate the contribution of these residues. On the other hand, our deep learning approach could not clearly detect the dynamic difference of N140, where hydrogen bonds were probably made with the ligand[50]. We assume that our approach could not detect N140 because of the minor change in the dynamics, considering that N140 is located in the interior side of the binding pocket hence its movement can be restricted. This is contrary to the detected residues (Trp81, Val87, and Leu92) that are exposed to the solvent (Supplementary Fig. 4). From the comparison to experimental literature, our methods can extract the important residues that change their dynamics significantly, e.g., residues exposed to the solvent. However, it is difficult to detect the residues with minor dynamics change, e.g., residues buried in the binding pocket.

Similarly to the comparison between apoprotein and ligand 3 systems (Fig. 3d) that were located in the extreme side of the embedding map, we compared dynamics of apoprotein to four other systems, in order to find a general trend that is observed in accordance with PC1. Supplementary Fig. 10 shows that more suppressed proteins were located with increasing PC1. This suggests that, while our methods can address high-dimensional data, the ligand-induced dynamics of BRD4 were largely characterized by its flexibility.

If dynamics is represented by the amplitude of fluctuation, the difference of dynamics is distinguished using RMSF equally to our deep learning approach. We attempted to characterize the dynamics with RMSF values and performed principal component analysis. Supplementary Fig. 11 shows that the PC1 obtained from RMSF calculation strongly correlated to the binding energies, which is comparable to our deep learning approach. However, for the other tested protein (see Discussion), the feature obtained from the RMSF-based dimension reduction correlated weakly to the binding energies. Therefore, we presume our deep learning approach is more generally applicable than the RMSF based method. This might be because the LDE can express more information such as the direction and temporal trend of the movements.

## Discussion

In this study, we have presented a deep learning approach that can determine the differences in protein behavior associated with the binding of different ligands. MD simulation data of apo and holoprotein systems were reduced to short-term LDE trajectories. Then, Wasserstein distances were calculated using DNN. Finally, the variables were extracted using dimension reduction methods for comparison of binding energies. For the BRD4 systems, there is a strong correlation between the ligand-induced dynamics and binding energies. Moreover, the characteristic short-term trajectories in the system were determined using $g(x)$ for the detection of key residues, which were also validated in experimental literature. To evaluate the generality of our approach, we also investigated systems of another protein, tyrosine phosphatase 1 B (PTP1B, Fig. 4a). Figure 4b illustrates clear separation of apoprotein systems from the cluster of holoproteins. This means that the ligand-induced dynamics are very different from the holoproteins but is similar among ligand-bound systems. The reason for the clustering of holoproteins could be that all the ligands with high affinity (more than 7.5 kcal/mol) interacted in a similar manner to PTP1B. In addition to just the separation, i.e., protein dynamics with high affinity ligand or no ligand, the PC1 correlated to the binding affinity with Pearson's coefficient 0.70 (Fig. 4c). This suggests that the PC1 distinguishes significantly favorable ligands from the others even within favorable ligand groups. We note that comparison with a broader range of

affinities would provide more apparent differences in protein dynamics and thus be more desirable for our method. For the two apoprotein systems, they were separated from each other. This might be because the pesudo-apo systems were not sufficiently relaxed in the simulations.

The strong correlation between the PC1 and binding energies in both BRD4 and PTP1B systems suggests that the relationship is somewhat general for other proteins and ligands. Firstly, the relation could not be restricted to specific types of proteins, as the natures of binding pockets in the BRD4 and PTP1B are different. The binding pocket is mainly hydrophobic in BRD4[48–50], while that is hydrophilic in PTP1B[43,44,51]. Secondly, the relation can hold true for both major and minor differences in tested ligands. The tested ligands in BRD4 are diverse in the mainframes, while those in PTP1B share the same frames but their terminals are different.

The main hyperparameters in our method are involved in MD simulations and LDE selection. First, MD simulations are required to be sufficiently long to sample the LDE. The number of data points exponentially reduced the error in the Wasserstein distances (Supplementary Fig. 12). Interestingly, a comparable result was obtained only from the fast 200 ns of simulation data (Supplementary Fig. 13), which suggests that sufficient LDE equilibrium can be obtained in the short simulations and minor differences in Wasserstein distances are removed in the embedding process. Second, the LDE time should be selected so that the resulting feature corresponds to a property of interest. In the case of binding affinity, the appropriate length was suggested to be that for the side-chain movement, although the results from different LDE time indicates robustness to the time selection (Supplementary Fig. 14). Thirdly, the selection of binding sites might be addressed by repeating the process multiple times. In the initial analysis, the input residues can be determined based on the distances to the ligand atoms. The proposed method can detect potentially important residues. In subsequent run, the extracted residues can be used to obtain a feature of the important protein dynamics. Finally, to interpret differences in dynamics, characteristic dynamics detected by $g(x)$ need to be expressed by the appropriate measurements. In the BRD4 systems, the characteristic behavior largely corresponded to the short-term RMSD. Depending on target proteins, the other measurements can be useful to explain characteristic behavior, such as the direction and temporal trend of movement.

In machine learning particularly for supervised learning, the required volume of training dataset and the calculation cost are the main concerns. In these points, our approach to extract principal components and predict ligand affinity is distinct from general supervised learning approaches. Our unsupervised learning approach essentially performs a dimension reduction method that reduces MD data from multiple systems to the principal components in a few dimensional space. This process does not use prior information on affinity. In contrast, with supervised learning, the algorithm learns patterns between input (e.g., sequence or coordinates) and output (e.g., affinity of ligand) from training data, thus requiring a known dataset whose amount matches the complexity of the training model. Recent machine learning models for affinity prediction are trained on the datasets that include at least thousands of protein–ligand complexes[9–14]. With regard to calculation cost, our approach involves a relatively expensive calculation of MD simulations and the DNNs that need to be calculated for a pair of systems. In contrast, supervised learning approaches provide output instantaneously once the model parameters are optimized. We could conclude that our approaches need more calculation and less known datasets compared to other machine learning methods. For the accuracy of prediction, our approach showed strong correlations in both target

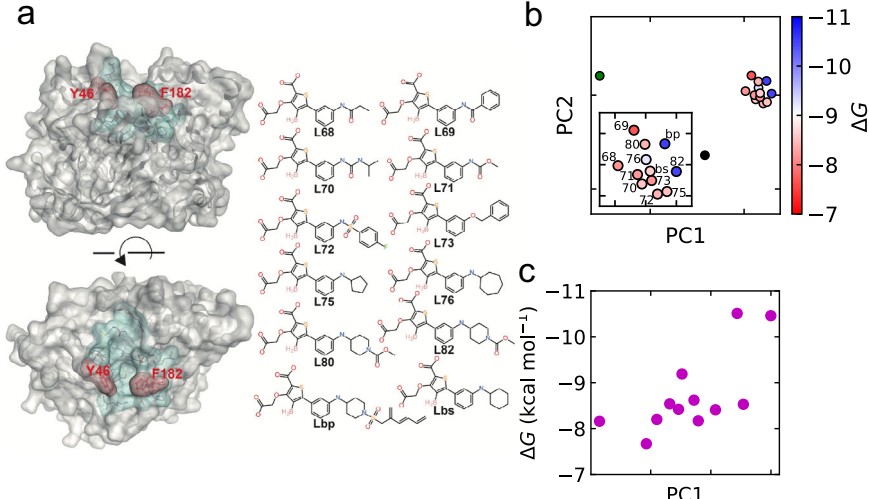

**Fig. 4 Ligand-induced dynamics in the phosphatase 1 B (PTP1B) systems. a** Molecular surface of PTP1B (Protein Data Bank ID: 1OEM) superimposed on ribbon diagram (left column). The green and red meshed molecular surfaces indicate the ligand binding site. The key residues (Tyr46 and Phe182, see Supplementary Fig. 19) are shown in red meshed molecular surfaces on ball and stick models with labels, respectively. These two residues were identified in experimental literature for the change of dynamics or the effect on the catalysis function[38,64,65]. Chemical structure of ligands (right column).
**b** Embedded points of the distance matrix. The holoprotein systems are colored according to the binding energies, and the crystal and pseudo apoprotein systems were colored in green and black, respectively. The binding energies were obtained from a previous computational research[4]. The inset shows the cluster of holoproteins. **c** Correlation between PC1 and the binding energies.

proteins, which is comparable to the other machine learning methods.

While our test cases were relatively rigid proteins that are widely used as benchmarks for free energy calculation, flexible proteins are interesting targets for further investigation. Dynamic properties may play a more important role in these systems. For a type of flexible protein, ligand-induced flexibility contributes to entropy gain in ligand binding, thus leading to higher affinity and longer residence of the ligands[28]. In fact, this was also seen in our cases in BRD4 with lig 4 (RVX-208) (Supplementary Fig. 10), where the binding is driven by entropy gain[50]. Furthermore, a similar approach could be used to study other protein–ligand binding events. For instance, it is interesting to evaluate the relationship between allosteric dynamics and ligand function, as has been done in a few previous studies[22,39]. Another potential application would be predicting the effects of protein mutations from the dynamics of the ligand. In this case, the relationship between the protein and ligand would be analyzed in a manner opposite to that employed in the present case, i.e., identical particles of the ligand would be used for LDE, while the protein molecules would vary slightly because of the mutation. We believe that the understanding of protein dynamics using ligand interactions will provide deeper insight into the function of ligands, and dynamics-based approaches would contribute to further developments in computational drug discovery.

## Method

**System setup and MD simulations**. For the BRD4 systems, the initial structures and topologies for proteins and ligands were considered according to a previous study[2]. The protein structures with and without ligands were solvated in a TIP3P[52] cubic box with a minimum distance of 1.0 nm. The systems were neutralized by adding $Na^+$ or $Cl^-$ ions. All-atom MD simulations of the systems were performed using GROMACS 2019.6[53]. The particle mesh Ewald method[54] was used to evaluate the electrostatic interactions with a cut-off radius of 1.2 nm, and van der Waals interactions were switched between 1.0 and 1.2 nm. The bonds with H atoms were constrained with LINCS[54] in the order of 4. For the prepared systems, energy minimization was carried out until the maximum force reduced to less than 10.0 kJ mol$^{-1}$, using the steepest descent method. Then restrained MD simulation was performed in a *NVT* ensemble at 300 K for 100 ps and subsequently in an *NPT* constant simulation at 1 bar for 100 ps. During both equilibration processes, position restraints were executed on the heavy atoms of the ligand and protein

atoms. The temperature and pressure were regulated using the velocity-rescaling method[55] and Berendsen methods[56], respectively. Finally, three individual production runs were performed for 400 ns in the *NPT* ensemble for each system with a random initial velocity generated to simulate different initial conditions. In the production runs, pressure was controled with Parrinello-Rahman pressure coupling method[57]. The trajectories were recorded every 2 ps.

For PTP1B systems, 12 complexes and two types of apoprotein systems were prepared for MD simulations (Supplementary Fig. 15). Ten complexes and pseudo apoprotein systems were constructed from the initial structures of PTP1B and the ligand used in the previous study by ref. [3]. In addition, another apoprotein was modeled from the crystal structure of PTP1B with no ligand (PDB ID: 1OEM) by homology modeling. Missing atoms were complemented using MOE[58], where we selected a structure without the α-helix. To distinguish between the two apoprotein systems, we denoted the apoprotein from a study by Song et al. as a pseudo-apo, which is originally complex and no ligand was added in our study. We called the apoprotein created from the crystal structure of the apoprotein as a crystal apoprotein. Proteins and water were parameterized by Amber ff14SB[59] and TIP3P[52], respectively. The parameters for the ligand were generated using GAFF[60] and parameter files in the study conducted by Song et al. The systems were solvated into a cubic box, with the thickness of the water shell set to 1.2 nm and neutralized with $Na^+$. MD simulations for PTP1B systems were performed similarly to the BRD4 systems, except for a few points. Energy minimization was performed for 10,000 steps, and the equilibration process was continued for 200 ps in both the *NVT* and *NPT* ensembles.

In the following analysis using machine learning, first 50 ns of trajectories were removed in both BRD4 and PTP1B systems. MD simulations were converged sufficiently in 50 ns (Supplementary Figs. 5, 6, 16, 17).

**Selection of the LDE**. Binding-site residues were mainly determined based on activity ratio that showed residue–ligand interaction based on the distance. We defined the activity ratio as $n/N$, where $n$ is the number of the trajectory frames in which the minimum heavy-atom distance between a residue and ligand is less than 0.5 nm, and $N$ is the total number of trajectory frames. We regarded residues with $n/N > 0.5$ to be in contact to the ligand. The activity ratio was calculated for each simulation in the first 200 ns, and residues were determined as the binding-site residues if any of the simulations identifies the residue–ligand contact. For BRD4 systems, we referred to the previous work by ref. [2] to further limit the number of residues. As a result, binding-site residues of BRD4 were 14 residues (Trp81, Pro82, Phe83, Val87, Leu92, Leu94, Tyr97, Cyc136, Tyr139, Asn140, Asp144, Asp145, Ile146, and Met149). Likewise, binding-site residues of PTP1B systems were 19 residues (Tyr46, Asp48, Val49, Lys120, Pro180, Asp181, Phe182, Gly183, Cys215, Ser216, Ala217, Ile219, Gly220, Arg221, Arg254, Met258, Gly259, Gln262, and Gln266). For the selection in PTP1B, we referred to the previous work by ref. [51].

From the trajectories of binding-site residues, rotation and translation were removed by fitting the trajectories to an identical structure in the backbone atoms

of the binding-site residues. This coordinate transformation is intended to best match coordinate systems between different systems of ligands.

Using the fitted trajectories, LDEs were generated. The LDE particles were the center of mass of the binding-site residues without hydrogen atoms, and the LDE time was 128 ps. We assume that the local dynamics ignores the average structure of binding-site residues by defining it as time-series displacements from a time step,

$$x = [r(t_0 + \Delta) - r(t_0), ..., r(t_0 + \delta) - r(t_0)] \tag{5}$$

where $r(t)$ is the positions of LDE particles at time $t$ in MD simulation, $\Delta$ is the time of LDE, and $\delta$ is the time interval of MD output. The local dynamics is implemented as a tensor in $(n, \Delta, d)$ dimension.

**Wasserstein distance between LDEs**. The Wasserstein distance between two probability distributions of LDEs is expressed as

$$W_{ij} = \sup_{\|f_{ij}\|_{L \leq 1}} \mathbb{E}_{x \sim y_i} \left[ f_{ij}(x) \right] - \mathbb{E}_{x \sim y_j} \left[ f_{ij}(x) \right] \tag{6}$$

where $i, j$ are the indexes for the systems, supremum is over all the 1-Lipschitz function $f$, $x$ is the short-term trajectory of the LDE, and $y_i$ is the LDE of system $i$. The advantages of Wasserstein distance over other measurements are (1) its applicability to high-dimensional data with affordable computation cost using DNN, (2) mathematical properties as a distance, which does not hold to divergence, and (3) no need for the prior assumption about the distribution[46].

To approximate the function $f_{ij}$, we used the DNN that was largely employed from the previous study by ref. [31]. The DNN was built using fully connected layers (Supplementary Fig. 18). The short-term trajectories $x$ were flatted and used as input for the DNN. The DNN had three hidden layers, whose number of output node was 2048. All the hidden layers used bias term and activation function of leaky rectified linear unit (LReLU). The output layer had one node without bias and activation function. The initial values of parameters were sampled from uniform distributions (mean = 0, deviation = $1/\sqrt{k}$), where $k$ is the number of the input features of each layer. The networks were implemented in pytorch[61].

In the optimization process, the loss function with gradient penalty[62] was minimized (see Supplementary Material for details). For each learning iteration, short-term trajectories were selected randomly by deciding the number of simulation and the initial step of the time sections. Model parameters were updated using Adam optimizer[63] (learning rate = 1e-4, beta 1 = 0.5, beta 2 = 0.9). The size of the minibatch was 64. The optimization process was performed for up to 500,000 steps per model, when the moving averages of DNN output over 10,000 steps converged. The mean value of the last 10,000 steps were used as the Wasserstein distances.

**Embedding of Wasserstein distances**. A Wasserstein distance matrix was embedded into vectors in low-dimensional space to satisfy the following equation,

$$p_0, p_1, ..., p_n = \arg\min_{p_0, p_1, ..., p_n} \sum_{i < j} (W_{ij} - \|p_i - p_j\|)^2 \tag{7}$$

where $p_i$ is an $n$-dimensional vector that corresponds to system $i$. The number of dimensions $n$ was set to three. The embedded vectors were optimized using simulated annealing and gradient descent (see Supplementary Material for details). Simulated annealing was employed to explore the global minimum and gradient descent for fast convergence. We iterated the embedding for multiple times and selected the best embedding with the minimum distance loss. Subsequently, principal component analysis was performed to the embedded vectors to obtain PC1 and PC2.

**Characteristic behavior analysis**. A function $g(x_i)$[31] represents the contribution of one short-term trajectory to the overall differences between two systems. For a LDE trajectory of system $i$ with referenced system $j$, the function $g(x)$ is defined as Eq. (2), and the equation is equivalent to

$$W_{ij} = \mathbb{E}_{x \sim y_i} \left[ g_{ij}(x_i) \right]. \tag{8}$$

The $g(x)$ quantitatively measures the uniqueness of one short-term trajectory as compared to the average dynamics of the other system. For instance, if a short-term trajectory in system $i$ has a small $g(x)$ when system $j$ is referenced, the short-term trajectory of system $i$ is similar to the average molecular behavior seen in system $j$, and basically vice versa. The $g(x_i)$ was calculated as the output of optimized DNNs when the inputs are the specific short-term trajectory from system $i$, and the average local dynamics of the other system $j$ in a pair (Supplementary Fig. 2). The $g(x)$ was sampled in every 64 ps of the MD trajectories.

**Statistics and reproducibility**. To obtain unbiased LDEs, MD simulations were performed three times with different initial velocities for each system. Because each simulation continued for 400 ns and the trajectories were recorded in every 2 ps, LDEs for 64 ps consisted of approximately 600,000 short-term trajectories for each system. We repeated calculations of Wasserstein distances several times and confirmed that they sufficiently converge regardless of the initial parameters of the

DNNs. Distance embedding was performed multiple times starting from randomly positioned embedded points. The sample size of short-term trajectories for $g(x)$ was 9375 for each ligand system.

**Reporting summary**. Further information on research design is available in the Nature Research Reporting Summary linked to this article.

## Data availability
Source data for figures are deposited at https://figshare.com/s/d3e3ff7aae04940adb78. Data of MD simulations are available upon reasonable request to the authors.

## Code availability
Analysis code is available upon reasonable request to the authors. The source codes for a sampling of short-term trajectories, DNNs to calculate Wasserstein distances, feature extraction, and g(x) calculation are essentially obtained from the authors of the work by ref. [31]. These codes are covered by their patent (Patent applicant: Keio University. Inventors: K. Yasuoka, D. Yuhara, K. Endo, and K. Tomobe. Application number: JP.2019048988.A. Status of application: published unexamined patent application).

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

## Author contributions

K.E., E.Y., and K.Y. conceptualized the research. I.Y. and Y.H. performed the simulations. I.Y and K.E. analyzed the data. I.Y., K.E., E.Y., Y.H., and K.Y. wrote and edited the manuscript.

## Competing interests

The authors declare no competing interests.
