## [Peer Review File · Communications Biology]

Reviewers' comments:

Reviewer #1 (Remarks to the Author):

The paper by Yasuda et al introduces an interesting Deep Learning method to predict/correlate the local dynamics of the binding sites of proteins to their affinity for ligands.

The method is based on MD simulations to investigate the dynamics and training and dimension reduction approaches to extract valuable information on affinity.

The paper is interesting although the concept that is presented is not new. Many papers in the past have tried to correlate dynamics with ligand effects.

In my opinion, the paper suffers from lack of clarity: it is very hard to follow, very technically involved. It would benefit from re-writing and simplification.

Also, I do not see how significant the correlation in fig. 4 could be for phosphatase, as all the binding values are concentrated in a single narrow region of space.

How generalizable is the use of the first principal component?

Also, is it possible to use other measures of distance?

I do not see the utility of the distance graphs (Fig 3d, Fig 4d) unless they are better explained.

In general, major revisions should be carried out on this paper before it is considered acceptable for publication.

Reviewer #2 (Remarks to the Author):

This interesting study proposed, based on molecular dynamics simulation data, an unsupervised deep learning method to measure the differences in protein dynamics at a ligand-binding site due to different bound ligands, and then used dimension reduction to extract a dynamic feature that is correlated to the binding affinities. Moreover, specific residues were predicted to play important roles in protein-ligand interactions. These results may be potentially useful for dynamics-based drug discovery.

Specific comments:

As a major caveat of deep learning, the 'learned' dynamic feature (PC1) lacks a meaningful/transparent explanation.

The authors introduced $g(x)$ to identify important residues but Fig 3d showed moving distances instead. How do they relate to each other?

It is unclear how the proposed deep learning based dynamic feature performs in comparison with alternative methods of detecting dynamic differences (e.g. using local RMSF etc) or direct calculation of binding energies (e.g. combining vdW and electrostatic energies). It is conceivable that these alternative (and inexpensive) methods may also obtain good (or even better) correlation with the binding free energy data.

The proposed important residues were not validated against experimental/functional literature.

In addition to ref[25], more references should be cited to account for the fast growing literature in applications of deep learning in MD simulation.

The architecture of DNN (depicted in Fig 1c) needs to be described in details so this method can be reproduced by interested readers. It will be even better if the code can be made accessible to public.

Reviewer #3 (Remarks to the Author):

In this manuscript Yasuda et al used a deep learning approach to investigate the relation between protein dynamics and protein-ligand binding affinities. Specifically, the dynamics of apo and holo BRD4 protein structures are simulated by classical molecular dynamics. These trajectories are simplified by local dynamics ensemble, and the simplified trajectories are used as input of a deep neural network, which computes the Wasserstein distances between trajectories. Then, the dimensionality of the data is reduced to relate the principal components with binding energies and to identify the key residues responsible for the binding process. Finally, the methodology is applied to the PTP1B protein also obtaining a good correlation between the binding free energy and the reduced-dimension protein dynamics.

Finding simple models that relate structural dynamics and binding affinities is crucial to develop new drugs in a very efficient way, and the present manuscript represents an important contribution in this direction. Moreover, the manuscript is very well written, the results are clearly presented and discussed, and the methods are technically appropriate. Therefore, this manuscript deserves to be published in Communication Biology after the following minor concerns are addressed:

1. Although the LDE and $g(x)$ function are explained in the methods section, I suggest adding a few lines with a general explanation along the results discussion in order to facilitate the reading of the paper to general readers who are not used to these properties.
2. The equilibration of the systems consist of a 100 ps trajectory in the NVT ensemble and 100 ps trajectory in the NPT ensemble (200 ps for the PTP1B protein). I found this equilibration very short. The authors should show that the protein structure is equilibrated and, if this is not the case, remove the snapshots from the production dynamics which are not equilibrated yet. In fact, the authors mention that 50 ns from the production was removed for the PTP1B protein, but this was not the case for the BRD4 system. Why did the authors choose 50 ns to be removed and why this was not done for the BRD4 protein? The equilibration of the protein could be discussed in more detail in the Supporting Information.
3. Although the deep neural network employed by the authors was already explained in a previous publication, it would be useful to have in the methods section a short explanation about the general features of the neural network.

I found two typos:

1. Line 42: "course-grained" should be "coarse-grained".
2. Line 54: "variable" should be "variables".

Reviewer #1 (Remarks to the Author):

The paper by Yasuda et al introduces an interesting Deep Learning method to predict/correlate the local dynamics of the binding sites of proteins to their affinity for ligands. The method is based on MD simulations to investigate the dynamics and training and dimension reduction approaches to extract valuable information on affinity. The paper is interesting although the concept that is presented is not new. Many papers in the past have tried to correlate dynamics with ligand effects.

Thank you for raising this point. Ligand effects on the dynamics of protein has been of great interest to protein–ligand complex study, and its relations to protein–ligand affinity have been discussed in some works. It has been experimentally investigated that the large conformational change at binding pocket such as transition between open and closed states of the binding pocket (Seo, *et al. Nat. Commun* 5 3724 (2014)), and ligand-induced local secondary structure change (Amaral et al, *Nat. Commun* 8 1 (2017)), is related to the ligand binding affinity.

On the other hand, in this study, we demonstrate that the dynamics in very short term at binding site correlates to the binding affinity. In this sense, our paper provides a new method that extract the effect of ligand binding only from subtle change included in the short-term trajectories, i.e. revealing the correlation between the protein’s local dynamics and ligand affinity. To clarify this point, we have revised the manuscript on page 3.

“It has been experimentally investigated that the large conformational change at the binding pocket such as transition between open and closed states of the binding pocket [37, 38], and ligand-induced local secondary structure change [28], is related to the ligand binding affinity. However, it is challenging to estimate the ligand binding affinity from just the subtle change included by the short-term MD trajectories [39].”

“In this study, we propose a novel method to predict binding energies from subtle change in proteins dynamics upon ligand binding, using a deep learning approach for MD data analysis [31].”

In my opinion, the paper suffers from lack of clarity: it is very hard to follow, very technically involved. It would benefit from re-writing and simplification.

We agree with the reviewer about the lack of clarity in the paper. For the simplification of the revised manuscript, we have added a new section to briefly explain our method (page 4 and 5, Unsupervised learning for ligand-induced dynamics), and the other technical details have been moved to the Method (pages 5-8). Furthermore, we have added step-by-step flowcharts (Supplementary Fig. 1 and 2).

Supplementary Figure 1. Workflow to extract feature of ligand-induced protein dynamics from MD simulations.

Supplementary Figure 2. Workflow to calculate function $g(\mathbf{x})$. S_i means system with ligand i . DNNs are optimized to calculate Wasserstein distances, and used to calculate the function $g(\mathbf{x})$.

Also, I do not see how significant the correlation in fig. 4 could be for phosphatase, as all the binding values are concentrated in a single narrow region of space.

Thank you for the comment. As the reviewer mentions, Fig. 4 shows a cluster of ligand-bound systems in PTP1B. This means the ligand-induced dynamics are very different from the holoprotein but is similar among the ligand-bound systems. The reason could be all the ligands tested in PTP1B have high affinities (more than 7.5 kcal/mol). The important point in the case of PTP1B is that our approach could detect ligand-induced dynamics by the PC1. In addition to just the separation, i.e. protein dynamics with high affinity ligand or no ligand, the PC1 strongly correlated to the binding affinity even within the favorable ligand groups. To demonstrate this point, we have updated the text on page 9:

“Fig. 4(b) illustrates clear separation of apoprotein systems from the cluster of holoproteins. This means that the ligand-induced dynamics are very different from the holoprotein but is similar among ligand-bound systems. The reason for the clustering of holoproteins could be that all the ligands with high affinity (more than 7.5 kcal/mol) interacted in a similar manner to PTP1B. The important point in the case of PTP1B is that our approach could detect ligand-induced dynamics by the PC1. In addition to just the separation, i.e. protein dynamics with high affinity ligand or no ligand, the PC1 strongly correlated to the binding affinity with Pearson's coefficient $r = 0.70$. This suggests that the PC1 distinguishes significantly favorable ligands from the others even within favorable ligand groups.”

How generalizable is the use of the first principal component?

Thank you for pointing this out. We believe that the first principal component (PC1) can generally extract important dynamics differences, and the correlation between binding affinity and PC1 is not limited to the specific cases in our study. For the tested proteins and ligands, the correlation of PC1 were verified on the different types of binding pockets and ligand groups. The pocket properties were

mainly hydrophobic with one hydrogen-bond point in BRD4 (Picaud et.al *PNAS* 110 19754 (2013)), while the binding pocket of PTP1B is mainly hydrophilic (Liu et. al *Sci. Rep* 4 1 (2014)). For ligands, different types of ligand groups were tested. The ligands in BRD4 systems did not share the mainframes, while those in PTP1B systems shared the identical mainframe. Despite these differences in the tested protein and ligand groups, PC1 was useful to estimate ligand affinity, hence the correlation is potentially general. To indicate this point clearly, we have revised the manuscript on page 9:

“The strong correlation between the PC1 and binding energies in both BRD4 and PTP1B systems suggests that the relationship is somewhat general for other proteins and ligands. Firstly, the relation could not be restricted to specific types of proteins, as the natures of binding pockets in the BRD4 and PTP1B are different. The binding pocket is mainly hydrophobic in BRD4 [48–50], while that is hydrophilic in PTP1B [43, 44, 51]. Secondly, the relation can hold true for both major and minor differences in tested ligands. The tested ligands in BRD4 are diverse in the mainframes, while those in PTP1B share the same frames but their terminals are different.”

Also, is it possible to use other measures of distance?

Thank you for raising this point. We used Wasserstein distance to measure differences between pairs of local dynamics ensembles. In principle, other measurements, e.g. KL divergence, JS divergence, can also be used. The reasons that we used Wasserstein distances are the followings:

- (1) Wasserstein distance is applicable to high dimension data with affordable computation cost using deep neural network.
- (2) mathematical properties as a distance, which does not hold to divergence.
- (3) no prior assumption about the distribution is required.

To make this point clear, we have revised the manuscript on page 13,

“The advantages of Wasserstein distance over other measurements are (1) its applicability to high dimensional data with affordable computation cost using DNN, (2) mathematical properties as a distance, which does not hold to divergence and (3) no need for the prior assumption about the distribution [46].”

I do not see the utility of the distance graphs (Fig 3d, Fig 4d) unless they are better explained.

We apologize for the insufficient explanations. The distance graph is now revised to the figure of short-term root-mean-square displacement (RMSD) (please see the definition on page 4). Briefly, the utility of the graph is to interpret characteristic dynamics obtained from the function $g(x)$ as the movement of each residue. Since multiple residues are included in LDE, the function $g(x)$ detects characteristic dynamics for the whole of these residues. Therefore, the function $g(x)$ cannot see the contributions per residue. To solve this, we calculate short-term RMSD for each residue. If the correspondence between characteristic dynamics and the short-term RMSD is clear, characteristic dynamics is interpreted as the flexibility of specific residues. We note that there is no theoretical connection between the function $g(x)$ and short-term RMSD. The revised manuscript includes the explanations in the section “Unsupervised learning for ligand-induced dynamics” on page 5.

“To clarify the specific dynamics contributing to the W_{ij} , we examine the residues included in the LDE by comparing the X_{ij}^C , X_{ij}^S groups, i.e. characteristic or non-characteristic dynamics of system i . We introduce a physical property to represent short-term trajectories and verify the property’s correspondence to the distinction of X_{ij}^C , X_{ij}^S groups. Here, short-term root-mean-square displacement (RMSD) was calculated”

“If the short-term RMSDs between X_{ij}^C and X_{ij}^S are distinct, i.e. $d_{ij}^C(n) \ll d_{ij}^S(n)$ or $d_{ij}^C(n) \gg d_{ij}^S(n)$, the dynamics of the residues contribute to W_{ij} .”

In general, major revisions should be carried out on this paper before it is considered acceptable for publication.

Thank you for the feedback. The manuscript have been revised by (1) adding a section “Unsupervised learning for ligand-induced dynamics” for the simplified explanation of our ML methods, (2) removing technical details from Result to Method section, (3) additional sentences to answer the reviewer’s pointing out.

Reviewer #2 (Remarks to the Author):

This interesting study proposed, based on molecular dynamics simulation data, an unsupervised deep learning method to measure the differences in protein dynamics at a ligand-binding site due to different bound ligands, and then used dimension reduction to extract a dynamic feature that is correlated to the binding affinities. Moreover, specific residues were predicted to play important roles in protein-ligand interactions. These results may be potentially useful for dynamics-based drug discovery.

We thank you the referee for the supporting assessment.

Specific comments:

As a major caveat of deep learning, the ‘learned’ dynamic feature (PC1) lacks a meaningful/transparent explanation.

Thank you for pointing this out. We would note that the PC1 is partially interpretable if we can reveal characteristic dynamics that contributed to the Wasserstein distance. This is because the PC1 is obtained from the Wasserstein distance. Here, we examined whether characteristic dynamics correspond to the short-term RMSD. If the correspondence between characteristic dynamics and the short-term RMSD is clear, characteristic dynamics is interpreted as the flexibility of specific residues, i.e. PC1 is interpretable. We compared the characteristic dynamics of apoprotein against holoprotein systems located in different PC1 values (Supplementary Fig. 10). The increasing PC1 is clearly related to the increasing movements of W81 in the characteristic dynamics of apoprotein.

To make this point clear, we have revised the manuscript on page 8:

“Similarly to the comparison between apoprotein and ligand 3 systems (Fig. 3d) that were located in the extreme side of the embedding map, we compared dynamics of apoprotein to four other systems, in order to find a general trend that is observed in accordance with PC1. Supplementary Fig. 10 shows that more suppressed proteins were located with increasing PC1. This suggests that, while our methods can address high-dimensional data, the ligand-induced dynamics of BRD4 were largely characterized by its flexibility.”

Supplementary Figure 10. Characteristic dynamics of BRD4 apoprotein, compared to holoprotein systems of increasing PC1 (system 3,5,7,10). Short-term RMSD was calculated for the holoprotein-like (low), apoprotein-like (high), and middle (mid) short-term dynamics. Orange number shows the ratio of the short-term RMSD of apoprotein-like dynamics to that of the holoprotein-like dynamics. The increasing PC1 mainly corresponds to lower mobility of W81.

The authors introduced $g(x)$ to identify important residues but Fig 3d showed moving distances instead. How do they relate to each other?

We apologize for the insufficient explanations. The distance graph is now revised to the figure of short-term root-mean-square displacement (RMSD) (please see the definition on page 4).

We note that there is no theoretical relation between short-term RMSD (moving distance) and $g(x)$. The function $g(x)$ detects characteristic dynamics for the whole of these residues, but it cannot show the contribution per the residue. On the other hand, short-term RMSD (on page 4) is calculated for the residue.

If the function $g(x)$ matches to short-term RMSD in residues, we can estimate that the residues contribute to the characteristic dynamics, i.e. interpreting characteristic dynamics. To clarify this point, we revised the manuscript on page 5:

“To clarify the specific dynamics contributing to the W_{ij} , we examine the residues included in the LDE by comparing the X_{ij}^C , X_{ij}^S groups, i.e. characteristic or non-characteristic dynamics of system i . We introduce a physical property to represent short-term trajectories and verify the property’s correspondence to the distinction of X_{ij}^C , X_{ij}^S groups. Here, short-term root-mean-square displacement (RMSD) was calculated”

“If the short-term RMSDs between X_{ij}^C and X_{ij}^S are distinct, i.e. $d_{ij}^C(n) \ll d_{ij}^S(n)$ or $d_{ij}^C(n) \gg d_{ij}^S(n)$, the dynamics of the residues contribute to W_{ij} .”

It is unclear how the proposed deep learning based dynamic feature performs in comparison with alternative methods of detecting dynamic differences (e.g. using local RMSF etc) or direct calculation of binding energies (e.g. combining vdW and electrostatic energies). It is conceivable that these alternative (and inexpensive) methods may also obtain good (or even better) correlation with the binding free energy data.

We thank the referee for the interesting suggestion. As a method using local RMSF, we introduce a local-RMSF based method. Briefly, the local-RMSF method calculates the RMSF of binding site residues. The RMSF values are used as features of systems, thus one system is represented by n features, where n is the number of residues in the binding site. The simple principal component analysis was applied to the n features to obtain a feature. Supplementary Fig. 11 compares the deep learning approach using local dynamics ensemble to the RMSF-base analysis. In BRD4 systems, the RMSF-base analysis was comparable to the deep learning approach, in terms of the correlation to binding energies. However, for PTP1B systems, our deep learning approach was better than the RMSF-base analysis. Therefore, we presume the deep learning approach is more general than the RMSF-based method. This might be because the local dynamics ensemble can express more information such as direction and temporal trend of the movement than the magnitude of fluctuation obtained by RMSF. The manuscript has been revised on page 8,

“If dynamics is represented by the amplitude of fluctuation, the difference of dynamics is distinguished using RMSF equally to our deep learning approach. We attempted to characterize the dynamics with RMSF values and performed principal component analysis. Supplementary Fig. 11 shows that the PC1 obtained from RMSF calculation strongly correlated to the binding energies, which is comparable to our deep learning approach. However, for the other tested protein (see Discussion), the feature obtained from the RMSF-based dimension reduction correlated weakly to the binding energies. Therefore, we presume our deep learning approach is more generally applicable than the RMSF based method. This might be because the LDE can express more information such as the direction and temporal trend of the movements.”

Supplementary Figure 11. Feature extraction using root mean square fluctuation (RMSF) at binding-site CAs and principal component analysis. The RMSF is calculated for each amino acid at the binding site in the last 150 ns of MD simulations (250–400 ns), and the RMSFs in three simulations for each system were averaged. Therefore, one system is represented by n features of the RMSF, where n is the number of the amino acids at the binding site. Subsequently, dimension reduction is performed using principal component (PC) analysis. (a) Map of PC1 and PC2 in BRD4 systems. (b) Map of PC1 and PC2 in PTP1B systems. (c) Correlation between the PC1 and the binding energies from free energy perturbation (FEP) in the BRD4 systems [5] ($r=0.87$). (d) Correlation between the PC1 and the binding energies from FEP in the PTP1B systems [6] ($r=0.32$).

The proposed important residues were not validated against experimental/functional literature.

We thank the referee for pointing this out. In the revised manuscript, we have compared the important residues detected in our methods to experimental studies that used crystallography and nuclear magnetic resonance (NMR) for BRD4. Ligand-induced dynamics changes on Trp81 were observed by NMR (Urlick et. al. *ACS Chem. Biol* 11 3154 (2016)). This study showed the dynamics change of Trp81 correlated to the ligand-binding affinity, in agreement with our result. In addition to Trp81, our result suggests that the other two residues, which have not been experimentally identified yet, have important roles in the ligand-binding. Previous simulation studies suggested that the residues at the binding site can make hydrophobic interactions (Ran et.al., *Mol. Biosyst* 11 1295 (2015), Wang et. al., *Chem. Phys. Lett* 736 136785 (2019)) and are expected to contribute to ligand binding. Although these residues are not clearly investigated in experiments, we hope for further experimental research to demonstrate the contribution of these residues.

On the other hand, our deep learning approach could not clearly detect the dynamical difference of N140, where hydrogen bonds were probably made with the ligands (Picaud et.al *PNAS* 110 19754

(2013). We assume that our approach could not detect N140 because of the minor dynamics changes, considering that N140 locates in the interior side of the binding pocket and the movements can be restricted. This is contrary to the detected residues (Trp 81, Val87, and Leu92) that are exposed to the solvent (Supplementary Fig. 4).

From the comparison to experimental literature, our methods can extract the important residues that highly changes their dynamics, e.g. residues exposed to the solvent. However, it is difficult to detect the residues with minor dynamics change, e.g. residues buried in the binding pocket.

The revised manuscript includes validation of important residues to experimental literature on page 7, 8.

“We estimate that the detected residues whose dynamics vastly changed by ligand have important roles in the ligand binding. To verify this, we refer to experimental and other computational literature on BRD4. Ligand-induced dynamics changes on Trp81 were observed by nuclear magnetic resonance [47]. This study showed the change in dynamics of Trp81 correlated to the ligand-binding affinity, in agreement with our result. In addition to Trp81, our result suggests that the other two residues, which have not been experimentally identified yet, have important roles in the ligand-binding. Previous simulation studies suggested that the residues at the binding site can make hydrophobic interactions [48, 49] and are expected to contribute to ligand binding. Although these two residues are not experimentally validated, we hope further experimental research will demonstrate the contribution of these residues. On the other hand, our deep learning approach could not clearly detect the dynamic difference of N140, where hydrogen bonds were probably made with the ligand [50]. We assume that our approach could not detect N140 because of the minor change in the dynamics, considering that N140 is located in the interior side of the binding pocket hence its movement can be restricted. This is contrary to the detected residues (Trp81, Val87, and Leu92) that are exposed to the solvent (Supplementary Fig. 4). From the comparison to experimental literature, our methods can extract the important residues that change their dynamics significantly, e.g. residues exposed to the solvent. However, it is difficult to detect the residues with minor dynamics change, e.g. residues buried in the binding pocket.”

In addition to ref [25], more references should be cited to account for the fast growing literature in applications of deep learning in MD simulation.

Thank you for the feedback. In the revised manuscript, we have cited more works for combined ML and MD. The manuscript have been revised on page 3,

“Recently, machine learning has been combined with MD to utilize the vast amount of MD data [29]. In particular, several machine-learning assisted methods succeeded in extracting important molecular dynamics [30–33], and were applied to complex biomolecules [34–36].”

The architecture of DNN (depicted in Fig 1c) needs to be described in details so this method can be reproduced by interested readers. It will be even better if the code can be made accessible to public.

We apologized the lack of the details of DNNs. The updated manuscript includes the architecture of the deep neural network in the Methods (page 13 and 14).

“To approximate the function f_{ij} , we used the DNN that was largely employed from the previous study by Endo et. al [31]. The DNN was built using fully connected layers (Supplementary Fig. 18). The short-term trajectories x were flattened and used as input for the DNN. The DNN had three hidden layers, whose number of output node was 2048. All the hidden layers used bias term and activation function of leaky rectified linear unit (LReLU).

The output layer had one node without bias and activation function. The initial values of parameters were sampled from uniform distributions (mean = 0, deviation = $1/\sqrt{k}$), where k is the number of the input features of each layer. The networks were implemented in pytorch [61].”

“In the optimization process, the loss function with gradient penalty [62] was minimized (see Supplementary Material for details). For each learning iteration, short-term trajectories were selected randomly by deciding the number of simulation and the initial step of the time sections. Model parameters were updated using Adam optimizer [63] (learning rate = $1e-4$, beta 1 = 0.5, beta 2 = 0.9). The size of the minibatch was 64. The optimization process was performed for up to 500,000 steps per model, when the moving averages of DNN output over 10,000 steps converged. The mean value of the last 10,000 steps were used as the Wasserstein distances.”

In addition, workflow for each process is described in Supplementary Fig. 1 and 2. We make our code publicly available upon reasonable request.

Reviewer #3 (Remarks to the Author):

In this manuscript Yasuda et al used a deep learning approach to investigate the relation between protein dynamics and protein-ligand binding affinities. Specifically, the dynamics of apo and holo BRD4 protein structures are simulated by classical molecular dynamics. These trajectories are simplified by local dynamics ensemble, and the simplified trajectories are used as input of a deep neural network, which computes the Wasserstein distances between trajectories. Then, the dimensionality of the data is reduced to relate the principal components with binding energies and to identify the key residues responsible for the binding process. Finally, the methodology is applied to the PTP1B protein also obtaining a good correlation between the binding free energy and the reduced-dimension protein dynamics.

Finding simple models that relate structural dynamics and binding affinities is crucial to develop new drugs in a very efficient way, and the present manuscript represents an important contribution in this direction. Moreover, the manuscript is very well written, the results are clearly presented and discussed, and the methods are technically appropriate. Therefore, this manuscript deserves to be published in Communication Biology after the following minor concerns are addressed:

We thank you the referee for the supporting assessment.

1. Although the LDE and $g(x)$ function are explained in the methods section, I suggest adding a few lines with a general explanation along the results discussion in order to facilitate the reading of the paper to general readers who are not used to these properties.

Thank you for the feedback. Because the previous version of the manuscript is technically involved without simple sentences for the explanations, we have added the result section “Unsupervised learning for ligand-induced dynamics” (pages 4 and 5) that contains the brief explanation of the concept of LDE and $g(x)$.

“Firstly, to represent protein dynamics, we use local dynamics ensemble (LDE) that is obtained from MD simulations (Figs 1a and 1b) [31]. The LDE is defined as an ensemble of short-term trajectories x of particles of interest.”

“we interpret the difference of protein dynamics using a function $g_{ij}(x_i)$ that shows how each short-term trajectory in system i differs from the average dynamics of system j (Figs 1c and Supplementary Fig. 2).”

2. The equilibration of the systems consist of a 100 ps trajectory in the NVT ensemble and 100 ps trajectory in the NPT ensemble (200 ps for the PTP1B protein). I found this equilibration very short. The authors should show that the protein structure is equilibrated and, if this is not the case, remove the snapshots from the production dynamics which are not equilibrated yet. In fact, the authors mention that 50 ns from the production was removed for the PTP1B protein, but this was not the case for the BRD4 system. Why did the authors choose 50 ns to be removed and why this was not done for the BRD4 protein? The equilibration of the protein could be discussed in more detail in the Supporting Information.

We thank the referee for pointing this out. In the previous version, we performed 200 ns production runs with NPT constant after pre-equilibration simulations in NPT and NVT constants with proteins restrained to stabilize the arrangement of water molecules. In this revision, we have extended the simulation time by 200 ns to verify the equilibration. Now the total time is 400 ns for each simulation. In the machine learning analysis, the first 50 ns production was removed for both BRD4 and PTP1B systems. The removed simulation was sufficiently long for protein equilibration judging from the RMSD of protein and ligand (Supplementary Fig. 5, 6, 16, and 17). To demonstrate this point, we have revised the manuscript on page 12,

“In the following analysis using machine learning, first 50 ns of trajectories were removed in both BRD4 and PTP1B systems. MD simulations were converged sufficiently in 50 ns (Supplementary Fig. 5, 6, 16, 17).”

Moreover, we have confirmed that the result does not change significantly if the ML analysis used the extended simulations (Supplementary Fig. 13). Thus, we presume that short simulation around 200 ns can produce equilibrated LDE. To demonstrate this point, we have revised the manuscript on page 9, *“Interestingly, a comparable result was obtained only from the fast 200 ns of simulation data (Supplementary Fig. 13), which suggests that sufficient LDE equilibrium can be obtained in the short simulations and minor differences in Wasserstein distances are removed in the embedding process.”*

3. Although the deep neural network employed by the authors was already explained in a previous publication, it would be useful to have in the methods section a short explanation about the general features of the neural network.

We apologize for the insufficient description. We have revised the Method section to briefly introduce neural networks. The manuscript was revised as noted in the last comment of the referee 2.

I found two typos:

1. Line 42: “course-grained” should be “coarse-grained”.
2. Line 54: “variable” should be “variables”.

We thank for the kind comment. We have fixed the typos.

REVIEWERS' COMMENTS:

Reviewer #1 (Remarks to the Author):

The authors have nicely and convincingly answered all the points raised. I think the paper is acceptable for publication.

Reviewer #2 (Remarks to the Author):

None.

Reviewer #3 (Remarks to the Author):

The authors have addressed all my concerns and they have significantly improved the manuscript by also addressing the comments of other reviewers. The methods section was extended and now it is more clear and complete than in the previous version. The explanation of the methods was the weakest point in the previous version, but in the revised one the authors have made an important effort to improve it. Therefore, in my opinion the paper is ready to be published.